# Denial of legal abortion in Nepal

**Mahesh C. Puri** [1]*, **Sarah Raifman**[2], **Sara Daniel**[3], **Sunita Karki**[1], **Dev Chandra Maharjan**[1], **Chris Ahlbach**[4], **Nadia Diamond-Smith**[2], **Diana Greene Foster**[5]

**1** Center for Research on Environment, Health and Population Activities (CREHPA), Kathmandu, Nepal, **2** Department of Epidemiology and Biostatistics, University of California, San Francisco, California, United States of America, **3** Department of Epidemiology, Johns Hopkins University, Baltimore, Maryland, United States of America, **4** School of Medicine, University of California San Francisco, San Francisco, California, United States of America, **5** Department of Obstetrics, Gynecology, Reproductive Sciences, Advancing New Standards in Reproductive Health, University of California, San Francisco, California, United States of America

\* mahesh@crehpa.org.np

## Abstract

### Introduction

In Nepal, abortion is legal on request through 12 weeks of pregnancy and up to 28 weeks for health and other reasons. Abortion is available at public facilities at no cost and by trained private providers. Yet, over half of abortions are provided outside this legal system. We sought to investigate the extent to which patients are denied an abortion at clinics legally able to provide services and factors associated with presenting late for care, being denied, and receiving an abortion after being denied.

### Methods

We used data from a prospective longitudinal study with 1835 women aged 15–45. Between April 2019 and December 2020, we recruited 1,835 women seeking abortions at 22 sites across Nepal, including those seeking care at any gestational age (n = 537) and then only those seeking care at or after 10 weeks of gestation or do not know their gestational age (n = 1,298). We conducted interviewer-led surveys with these women at the time they were seeking abortion service (n = 1,835), at six weeks after abortion-seeking (n = 1523) and six-month intervals for three years. Using descriptive and multivariable logistic regression models, we examined factors associated with presenting for abortion before versus after 10 weeks gestation, with receiving versus being denied an abortion, and with continuing the pregnancy after being denied care. We also described reasons for the denial of care and how and where participants sought abortion care subsequent to being denied. Mixed-effects models was used to accounting clustering effect at the facility level.

### Results

Among those recruited when eligibility included seeking abortion at any gestational age, four in ten women sought abortion care beyond 10 weeks or did not know their gestation and just over one in ten was denied care. Of the full sample, 73% were at or beyond 10 weeks gestation, 44% were denied care, and 60% of those denied continued to seek care

**Data Availability Statement:** Data used in the preparation of this paper is uploaded at https://dataverse.harvard.edu/dataset.xhtml?persistentId=doi:10.7910/DVN/HMOWCA.

**Funding:** This study was supported by the National Institute of Health (Grant number: A133916), and the David and Lucile Packard Foundation (Grant Number 132968) to Ms. Diana Greene Foster, at the University of California, San Francisco. The funders had no role in study design, data collection and analysis, decision to publish, or preparation of the manuscript.

**Competing interests:** The authors have declared that no competing interests exist.

after denial. Nearly three-quarters of those denied care were legally eligible for abortion, based on their gestation and pre-existing conditions. Women with lower socioeconomic status, including those who were younger, less educated, and less wealthy, were more likely to present later for abortion, more likely to be turned away, and more likely to continue the pregnancy after denial of care.

## Conclusion

Denial of legal abortion care in Nepal is common, particularly among those with fewer resources. The majority of those denied in the sample should have been able to obtain care according to Nepal's abortion law. Abortion denial could have significant potential implications for the health and well-being of women and their families in Nepal.

## Background

Providing women access to safe and legal abortion services is essential to realizing and protecting their fundamental human rights. These services enable women to control their fertility, protect their health, and ensure the wellbeing of their families [1,2]. About 59% of women of reproductive age live in countries where legal abortion is available within certain grounds [3], and women who seek care beyond these grounds are denied services. The few studies focused on measuring abortion denial in countries outside the United States have found that between 2% and 45% of women are turned away when seeking legal abortion services (2% in Columbia, 26% in Nepal to 45% in South Africa), with many women seeking unsafe abortions elsewhere subsequent to denial and others reporting that they anticipated additional hardships if they carried the unwanted pregnancy to term [4–9].

Abortion has been conditionally legal in Nepal since 2002 under broad criteria. The legal criteria were revised in 2018 with the enactment of a new law, the Safe Motherhood and Reproductive Health Rights Act, which permits abortion up to 12 weeks gestation on request and up to 28 weeks gestation if the pregnancy resulted from rape or incest, if the woman suffers from HIV or other similar types of incurable diseases, or if the pregnant woman has specific mental health conditions. Abortion is also permitted if the pregnancy poses a danger to the woman's life or her physical or mental health, or if there is a fetal abnormality; for these cases, an approved medical practitioner's recommendation is required [10]. Having an abortion in an effort to have a child of a specific sex (sex-selective abortion) is not permitted.

During the past 20 years, the Nepal Ministry of Health and Population has developed strategies for implementing the law and expanding access to safe and legal abortion services. These strategies include training clinicians to perform abortions, providing them with the necessary equipment, and certifying health facilities [11,12]. The number of certified health facilities for abortion in Nepal has steadily expanded since 2004; by 2020, about 4,521 clinicians were trained and 1,516 facilities were certified [13]. Since 2008, nurses in addition to physicians have been eligible to receive training in manual vacuum aspiration up to 8 weeks gestation. Second- trimester abortion training for physicians began in 2007, and by 2020, 22 hospitals were providing second-trimester abortion in the country [13]. In 2009, medical abortion within 9 weeks gestation was introduced initially as a pilot program in six districts and has been gradually scaled up to the entire country.

Despite concerted efforts to expand legal and safe termination services, these services remain inaccessible for many women in Nepal, especially low-income, socially marginalized,

and geographically isolated women [6]. Of the estimated 323,200 abortions carried out in Nepal in 2014, over half (58%) were provided illegally [14]. A recent modelling study found that a 10% shift in abortion from safe to unsafe would result in 14,500 additional unsafe abortions annually in Nepal [15]. Lack of awareness about the legal provisions for abortion, availability, location, and costs of services, as well as access to transport to approved facilities, prevent many women from accessing obtaining safe and legal abortion services [12,16]. Other cultural barriers, including a lack of autonomy in reproductive decision-making due to patriarchal norms about family planning and religious beliefs, also limit women's access to legal abortion services [12,16].

Although there is sufficient evidence that mid-level providers such as nurses and midwives can provide medical abortion as safely and effectively as physicians, the government has been slow to scale-up training such providers, a move which could greatly expand the numbers and locations of abortion providers [17]. Only 38% of all public facilities permitted to provide abortion services reported offering these services in 2014 [18]. Furthermore, at that time, less than half of all public facilities in Nepal that are permitted to provide post-abortion care reported doing so [18]. The covid-19 pandemic may have further affected the availability and quality of abortion services [15].

Fees for abortion services in private facilities are not regulated and are often prohibitively expensive [11]. The 2015 government policy of providing cost-free abortion in public facilities is an important step in addressing cost barriers. However, anecdotal evidence and qualitative data suggest this policy is unevenly enforced [6]. A previous study that collected data from providers in Nepal suggested that many women who should legally qualify for free public services are denied care, even those who are under the 12-week gestational age limit for termination on request [19]. Many providers do not correctly screen for eligibility for services beyond 12 weeks gestation and most do not know the criteria for services above this limit [19].

An exploratory qualitative study on the denial of abortion services in Nepal showed that one-quarter of women did not receive legal abortion services on the day of their visit [6,9], most commonly because they were beyond 12 weeks gestational age, seeking a sex-selective abortion, or they had a possible health contraindication [9,19]. Although previous studies provide important information about the experiences of women who seek abortion services in Nepal, there is a need for systematic, quantitative evidence on the extent of abortion denial, including who is and is not able to receive a legal abortion. Such data could help to identify strategies to improve access to abortion services in Nepal and similar settings where abortion is legal. In this paper, we present data from a longitudinal study of women who sought legal abortion services in Nepal in 2019 and 2020 and explore the extent of legal abortion denial and factors associated with denial in Nepal.

## Data and methods

The Nepal Turnaway Study is a prospective longitudinal study to evaluate the effects of receiving versus being denied legal abortion in Nepal on maternal mental and physical health as well as the health and socioeconomic consequences for women and their families.

Between April 2019 and December 2020, we recruited and consented 1,835 women seeking abortions at 22 sites across Nepal, including those seeking care at any gestational age (n = 537, April-May 2019) and then only those seeking care at or after 10 weeks of gestation (n = 1,298, May 2019-December 2020). We conducted interviewer-led surveys with participants at their home or other chosen location at six weeks after abortion-seeking and every six months for three years. We began recruitment in April 2019 at 14 diverse public and private/non-profit facilities (one of each type in each of 7 provinces), selected randomly with chance of selection

proportionate to their client volume from a list of certified abortion facilities that provided 60 or more abortions per year in 2016–2017. The facilities included in the sampling frame provided 92% of legal abortion services in the country in that time period. Due to the low volume of eligible study participants at some of these initial sites, we replaced seven of the original 14 sites and added one additional site in mid-2019, using the same sampling strategy based on 2016–2017 service data.

Women over the age of 15 seeking abortion care, and living in Nepal were eligible for study participation in the first month of recruitment (mid-April to mid-May 2019). From mid-May 2019 to December 2020 (excluding a 3-month pause in recruitment due to Covid-19 travel restrictions), we restricted study eligibility to women who presented for care at or beyond 10 weeks gestation or who did not know their gestational age in order to collect a sufficient sample of women who would likely be turned away.

All patients presenting for care were screened for study eligibility by a point person at the facility. These point people–doctors, nurses, counselors, or receptionists–completed an eligibility form for every woman seeking an abortion over the entire study period. The form recorded the woman's age, estimated gestation, provider assessment of eligibility for abortion, and reason for ineligibility, if relevant. If a woman was eligible for the study, the point person at the facility referred her to speak with a trained research staff member who was stationed in a private room at each clinic. The research staff member confirmed study eligibility, obtained written informed consent (a thumbprint was obtained for women unable to sign), conducted the baseline survey in the clinic using a tablet, and uploaded survey answers to a secure web-based storage platform. In the case of minors under 18 years of age, participants provided assent for participation and interviewers obtained consent from one biological parent. Interviewers contacted all participants six weeks after recruitment and every six months thereafter for the next three years. The interviewer conducted surveys in Nepali, Maithali, Tharu, Bhojpuri, or Hindi, according to the participant's preference. Interviews took roughly 45 minutes on average. Each participant received financial compensation equivalent to about $4 USD for the baseline and each subsequent interview. The University of California, San Francisco Human Research Protection Program and the Nepal Health Research Council provided ethical review and approval.

The present study used a cross-sectional analysis of data collected from the eligibility forms, baseline interviews, and 6-week interviews. In these interviews, we collected data on basic demographic and socioeconomic characteristics such as age, marital status, number of children, years of education, whether the woman worked outside the home, and caste/ethnicity. Consistent with the Nepal Demographic Health Survey methodology, we calculated wealth quintiles using principal component analysis of more than 40 household asset items.

To understand the reasons for abortion-seeking, we asked the participant whether their pregnancy was a result of rape or incest and whether a doctor, nurse, or other health worker told them that their health or life was at risk because of the pregnancy or that the baby might have severe health problems. We asked whether they experienced any of 11 adverse feelings in the weeks since they became pregnant (severe difficulty falling asleep; always sleepy or falls asleep all the time; lethargic or less energetic; guilty or worthless all the time; feeling that life has become meaningless and unsupported; problems concentrating, carefully thinking, or making decisions; excited, restless or irritated; hesitation participating in recreational activities; unable to take care of other children financially, mentally and physically; believes the baby will affect her education and professional career; and believes the pregnancy is the result of an extramarital affair). Those experiencing three or more are legally eligible to obtain an abortion for mental health reasons in Nepal. We asked participants the primary reason they decided to have an abortion as an open-ended question. Interviewers recorded responses into 11

categories based on previous research or as open text for other answers: have enough children; can't afford additional children; youngest child is small/breastfeeding; I am too young; wanted a child of a different sex; husband was away when conceived; family problems; studying; health problems; husband wants me to have an abortion; and family members want me to have an abortion.

To understand the timing of abortion-seeking, in the baseline interview we asked when the participant first discovered she was pregnant, whether she made any attempts to end the pregnancy prior to presenting at the recruitment clinic, and how long it took to get to the abortion clinic. In the six-week interview, we asked whether the participant was aware that abortion was legal in Nepal and whether she had received the abortion from the recruitment facility or had been turned away. If she did not receive an abortion from the recruitment facility, we asked whether she continued to seek care elsewhere and whether she was still pregnant.

In this paper, we examined the factors associated with presenting for abortion before versus after 10 weeks gestation as one measure of access to abortion services. we examined differences by gestational age in who received or were denied an abortion and their reasons for the denial, among those who completed a 6-week or subsequent follow-up survey. We also examined factors associated with denial compared to receipt of abortion and factors associated with continuing the pregnancy after being denied. To do this, we used bivariable and multivariable mixed-effects logistic models accounting for clustering at the facility level, given that patient characteristics and service provision protocols (including denial of care) may be more similar within a given facility. We include descriptive characteristics and factors associated with eligibility for and access to abortion. Timing of discovery of pregnancy and previous abortion attempts were not included as these are on the causal pathway to late presentation for abortion. All analysis were done in Stata 15.1.

## Results

Between April 16, 2019 and December 31, 2020, 8,856 women sought an abortion at one of the 22 participants recruitment sites. Of these, 1,925 (21.7%) were eligible for the study (six participants were removed from the sample during analysis after it was determined they were not pregnant or their period returned soon after the initial clinic visit) and 1,835 (95.3% of eligible women) consented to participate and completed a baseline interview. 1,668 (90.9% of those who enrolled) completed at least one subsequent interview.

### Presenting for abortion at or beyond 10 weeks gestation

Based on the findings from the first month of recruitment, during which period we recruited a representative sample of all women seeking abortions in Nepal, 40% presented beyond 10 weeks gestation or did not know their gestational age. During the full recruitment period (one month of recruiting everyone followed by 19 months of recruiting only those beyond 10 weeks gestation or who were denied abortions for any reason), nearly three quarters (73%) were at or beyond 10 weeks gestation.

Participants who were young, non-married, less educated, less wealthy, and from the Dalit caste were more likely to present for an abortion beyond 10 weeks (Table 1). Some logistical factors also increased the chance participants presented at or beyond 10 weeks, such as traveling more than 3 hours to get to the clinic, discovering pregnancy after six weeks gestation, and having previously attempted to terminate the pregnancy elsewhere. Those who were aware that abortion is legal or who had a previous abortion were less likely to present beyond 10 weeks gestation (Table 1).

**Table 1. Characteristics of women seeking abortion by gestational age among those who completed the baseline survey.**

| | | | Gestational age | | |
|---|---|---|---|---|---|
| | Total (N = 1835) | | <10 weeks gestation (n = 483) | >10 weeks or don't know (n = 1,352) | |
| | n | % | % | % | P values from mixed effects |
| **Women's age (in years)** | | | | | |
| <24 | 627 | 34 | 21 | 79 | 0.003 |
| 25–29 | 528 | 29 | 24 | 76 | 0.023 |
| 30–34 | 392 | 21 | 32 | 68 | 0.626 |
| 35–45 | 288 | 16 | 34 | 66 | ref |
| **Marital status** | | | | | |
| Single/divorce/widow | 61 | 3 | 13 | 87 | ref |
| Married | 1,764 | 97 | 27 | 73 | 0.012 |
| **Number of children** | | | | | |
| No children | 256 | 14 | 22 | 78 | ref |
| 1 | 548 | 30 | 26 | 74 | 0.078 |
| 2 | 652 | 36 | 30 | 70 | 0.002 |
| 3 and more | 379 | 21 | 24 | 76 | 0.139 |
| **Level of education** | | | | | |
| None / non-formal | 293 | 16 | 22 | 78 | ref |
| Primary (1–5 years) | 280 | 15 | 23 | 78 | 0.623 |
| Secondary (6–12 years) | 1,128 | 62 | 27 | 73 | 0.208 |
| More than secondary | 124 | 7 | 35 | 65 | 0.005 |
| **Employment status** | | | | | |
| No | 845 | 46 | 26 | 74 | ref |
| Yes | 981 | 54 | 27 | 73 | 0.852 |
| **Caste/Ethnicity** | | | | | |
| Brahmin/Chhetri | 716 | 39 | 27 | 73 | ref |
| Hill Janajati | 432 | 24 | 28 | 72 | 0.090 |
| Dalit | 238 | 13 | 20 | 80 | 0.016 |
| Terai Janajai | 392 | 21 | 28 | 72 | 0.762 |
| Others | 48 | 3 | 6 | 94 | 0.055 |
| **Facility Type** | | | | | |
| Public | 525 | 29 | 23 | 77 | ref |
| Private/NGO clinic | 1,310 | 71 | 28 | 72 | 0.633 |
| **Travel Time to clinic** | | | | | |
| Up to 1/2 hour | 557 | 31 | 41 | 59 | ref |
| >1/2 to 1 hour | 419 | 23 | 25 | 75 | 0.000 |
| >1 to 3 hours | 435 | 24 | 20 | 80 | 0.000 |
| >3 to 24 hours | 396 | 22 | 13 | 87 | 0.000 |
| **Discovered Pregnancy** | | | | | |
| Before 6 weeks | 1,264 | 71 | 34 | 66 | ref |
| At or after 6 weeks | 515 | 29 | 8 | 92 | 0.000 |
| **Prior attempts at abortion for this pregnancy** | | | | | |
| No | 1,601 | 88 | 27 | 73 | ref |
| Yes | 225 | 12 | 17 | 83 | 0.012 |
| **Completed 6-week survey** | 1668 | 100 | 27 | 73 | |
| **Aware of legal abortion provision** | | | | | |
| No | 758 | 45 | 19 | 81 | ref |

*(Continued)*

**Table 1.** (Continued)

| | | | Gestational age | | |
|---|---|---|---|---|---|
| | Total (N = 1835) | | <10 weeks gestation (n = 483) | >10 weeks or don't know (n = 1,352) | |
| | n | % | % | % | P values from mixed effects |
| Yes | 848 | 51 | 34 | 66 | 0.000 |
| To some extent | 61 | 4 | 31 | 69 | 0.426 |
| **Quintiles of wealth** | | | | | |
| 1-lowest | 329 | 20 | 14 | 86 | ref |
| 2 | 328 | 20 | 22 | 78 | 0.017 |
| 3 | 327 | 20 | 26 | 74 | 0.002 |
| 4 | 326 | 20 | 32 | 68 | 0.000 |
| 5-highest | 327 | 20 | 39 | 61 | 0.000 |
| **Previous abortion** | | | | | |
| No | 1,007 | 79 | 23 | 77 | ref |
| Yes | 215 | 21 | 40 | 60 | 0.000 |

The most commonly reported reasons for seeking abortion were already having enough children (46%), their youngest child was small or still breastfeeding (22%), and unable to afford another child (15%). Most reasons for seeking abortion did not vary substantially by gestational age at the time of care-seeking, with the exception of the following: those below 10 weeks were more likely to report having enough children (58% vs 41%, p<0.001) and having health problems (18% vs 13%, p<0.001) and less likely to report wanting a child of a different sex (0% vs 13%, p<0.001) or being too young (3% vs 5%, p = 0.008) (Table 2).

## Denial of abortion

During the month where recruitment reflected the population seeking abortion nationally, 11% of those seeking care were denied, according to participant reports at the 6-week interview. In the larger sample including participants from the full recruitment period, 736 (44%) were denied care at the recruitment facility. Of those who participated in the 6-week interview, 855 (51%)

**Table 2. Reasons for abortion by gestational age at time of abortion seeking.**

| Reasons for abortion | Total (N = 1,835) | <10 weeks gestation (n = 483) | > = 10 weeks gestation/ don't know (n = 1,352) | | P values from mixed effects |
|---|---|---|---|---|---|
| | % | % | % | | |
| Have enough children | 46 | 58 | 41 | * | 0.000 |
| Youngest child small/breast feeding | 22 | 18 | 23 | | 0.085 |
| Can't afford additional children | 15 | 14 | 16 | | 0.580 |
| Health problems | 14 | 18 | 13 | | 0.000 |
| Family problems | 10 | 10 | 10 | | 0.134 |
| Wanted a child of a different sex | 9 | 0 | 13 | | 0.000 |
| Studying | 7 | 9 | 7 | | 0.822 |
| I am too young | 5 | 3 | 5 | | 0.008 |
| Husband wants me to have an abortion | 2 | 2 | 2 | | 0.784 |
| Family members want me to have an abortion | 1 | 1 | 1 | | 0.668 |
| Husband away when conceived | 1 | 1 | 1 | | 0.737 |
| Other | 11 | 8 | 12 | | 0.139 |

**Table 3. Reasons for denial of abortion by gestational age at time of abortion seeking.**

| | <10 weeks gestation (n = 58) | >10 weeks/ don't know (n = 683) | Total (n = 741) | P values from mixed-effects |
|---|---|---|---|---|
| | % | % | % | |
| Provider said I was too far along | 7 | 84 | 78 | 0.000 |
| I was not sure I wanted an abortion | 14 | 5 | 6 | 0.014 |
| Provider said they don't do abortion | 3 | 6 | 6 | 0.374 |
| Provider not available | 26 | 4 | 5 | 0.000 |
| I didn't have money | 3 | 4 | 4 | 0.972 |
| Provider said I have other medical problems so they couldn't do the abortion | 14 | 3 | 4 | 0.000 |
| Pregnancy too early | 19 | 2 | 3 | 0.000 |

received an abortion at the recruitment facility the day of study enrollment; 72 (4%) were denied but received an abortion from that clinic at a later date; 477 (29%) were denied and received an abortion elsewhere or had a miscarriage or stillbirth, and 259 (15%) were denied and still pregnant at their 6-week interview (not shown in tables). Women presenting below 10 weeks gestation were much less likely to be denied an abortion at the recruitment facility than those at or above 10 weeks (13% vs 56%, p<0.001). The most common reason for denial of abortion among those at or above 10 weeks was advanced gestation (84%). For those under 10 weeks, common reasons for denial included lack of provider availability (26%), early pregnancy (19%), medical contraindications (14%), or the patient was not sure about wanting an abortion (14%). (Table 3).

Based on gestation and responses to questions about reasons for abortion, we estimate 97% of those who received an abortion and 78% of those who were denied an abortion were legally eligible for the procedure (Table 4). Four percent of those seeking abortions were not legally eligible because their only reason for abortion was to select the sex of the fetus; most of these patients were denied care at the recruitment facility. Seven percent were beyond 12 weeks gestation (the legal limit for abortion on request in Nepal) and did not have a condition that would have allowed the procedure; most of these patients were also denied care. However, there were others who were legally eligible who were also denied abortions. Of those denied an

**Table 4. Denial of abortion by legal status.**

| | Received Abortion (n = 849) | Total Denied (n = 674) | Denied abortion but no longer pregnant (n = 444) | Denied and carrying to term (n = 230) | Total (n = 1,523) |
|---|---|---|---|---|---|
| | % | % | % | % | % |
| **Legally eligible <12 weeks** | 88 | 53 | 58 | 45 | 73 |
| **Legally eligible beyond 12 weeks*** | 9 | 24 | 23 | 27 | 16 |
| 3+ mental health symptoms | 8 | 23 | 21 | 28 | 15 |
| Physical health reasons | 2 | 3 | 4 | 2 | 2 |
| Fetal diagnosis | 4 | 3 | 4 | 0 | 3 |
| Rape/incest | 0 | 0 | 0 | 0 | 0 |
| **Not legally eligible (sex selection)** | 1 | 9 | 9 | 8 | 4 |
| **Not legally eligible beyond 12 weeks** | 2 | 13 | 10 | 19 | 7 |

Note: Percentages add to more than 100 because some women qualify for abortions past 12 weeks on multiple grounds.

abortion, half (53%) were legally eligible because they had a gestation below 12 weeks of pregnancy, and another quarter were beyond 12 weeks but had a condition for which abortion is permitted under the law. Of those who were denied and should have qualified, many had three or more mental health conditions (94%), physical health reasons (12%), a fetal diagnosis (11%), or were seeking an abortion after rape or incest (<1%).

## Subsequent abortion-seeking after denial

Of those denied at the recruitment facility, 442 (60%) reported seeking subsequent abortion care, including 14 (2%) who reported two subsequent abortion attempts. Most of the total subsequent abortion attempts (n = 456) involved seeking care at a facility (89%, n = 407) and most resulted in the woman obtaining a procedure (294, 64%; not shown in tables). Participants sought care after denial at private clinics (43%), public hospitals (29%), and private hospitals (15%); others went to hospitals or clinics in India (3%), primary health centers (1%), and pharmacies (1%). About one quarter of all subsequent abortion attempts were reported to involve taking medicines, tablets, or pills and some other method without a procedure (23%, n = 106). A small proportion (6%, n = 24) of the 456 subsequent abortion attempts involved facility care other than a procedure or pills (such as physical exam, counseling, ultrasound, or referral) and 7% (n = 29) received unknown care at a facility. In two cases, participants reported that they drank home remedies to terminate the pregnancy.

In multivariate analyses, among those presenting for abortion at or after 10 weeks, denial was more likely for those who were seeking abortion for reasons of sex selection (aOR 9.39, 95% CI: 3.9,22.58), under age 25 (OR 1.78, 95% CI: 1.02,3.10), unmarried (OR 2.97, 95% CI: 1.22,7.23), in the lowest quintile of wealth (OR 1.78, 95% CI: 1.00,3.15), not working outside the home (OR 1.51, 955 CI: 1.08,2.10), and unaware of the legal status of abortion in Nepal (OR 1.40, 95% CI: 1.01,1.92) (Table 5). Women with no children (OR 0.41, 95% CI: 0.21–0.82) or who reported previous abortions (OR 0.68, 95% CI: 0.47,0.99) and those who were seeking abortion for fetal anomaly diagnosis (OR 0.51, 95% CI: 0.28, 0.93) were less likely than others to be turned away.

After the denial of abortion, we see patterns of social disadvantage in who was unable to get an abortion elsewhere. Young women were much more likely than those 30 or older to still be pregnant at six weeks (OR 2.62 [1.26, 5.42]) for those under 25 and OR 2.77 [1.45, 5.28] for those 25–29. The same was true for those with lower levels of education (OR 3.13 [1.04, 9.43] those with informal or no education and OR 3.77 [1.31, 10.85] those with only primary education), those who did not work outside the home (OR 1.72, 95% CI: 1.11, 2.66), who were in the Dalit caste (OR 2.01, 95% CI: 1.14, 3.53), and who had among the lower levels of wealth (OR 4.38 [1.99, 9.64] for the lowest quintile, OR 2.32 [1.11, 4.84] for the second-lowest, and OR 2.07 [1.03, 4.19] for the middle quintile). Unmarried women were much less likely than married women to carry the pregnancy to term after being denied an abortion (OR 0.08, 95% CI: 0.02, 0.32).

## Discussion

A consistent pattern of differences emerged between women who presented early for abortion services compared to those who presented later, between women who received compared to those who were denied their abortions, and between those who got an abortion elsewhere after being denied compared to those who carried the pregnancy to term. Younger women with lower wealth and education levels and those of the Dalit caste were at increased risk of presenting for abortion later in pregnancy, being denied care, and carrying the pregnancy to term after denial. One explanation for this is that these women were likely to be more disadvantaged

**Table 5. Predictors of denial of abortion and of carrying the pregnancy to term among women over 10 weeks of pregnancy.**

| | Predictors of Denial | | | Predictors of Carrying Pregnancy to Term after Denial | | |
|---|---|---|---|---|---|---|
| | Adjusted Odds Ratio | P value | 95% Confidence Interval | Adjusted Odds Ratio | P value | 95% Confidence Interval |
| **Women's age (in years)** | | | | | | |
| <24 | 1.78 | 0.042 | [1.02, 3.10] | 2.62 | 0.010 | [1.26, 5.42] |
| 25–29 | 1.29 | 0.297 | [0.80, 2.09] | 2.77 | 0.002 | [1.45, 5.28] |
| 30–34 | 1.03 | 0.891 | [0.64, 1.68] | 1.31 | 0.434 | [0.67, 2.58] |
| 35–45 | Ref | | | Ref | | |
| **Marital status** | | | | | | |
| Single/divorce/widow | 2.97 | 0.016 | [1.22, 7.23] | 0.08 | 0.000 | [0.02, 0.32] |
| Married | Ref | | | Ref | | |
| **Number of children** | | | | | | |
| No children | 0.41 | 0.012 | [0.21, 0.82] | 2.28 | 0.097 | [0.86, 6.03] |
| 1 | 0.63 | 0.075 | [0.38, 1.05] | 1.22 | 0.544 | [0.64, 2.34] |
| 2 | 0.92 | 0.693 | [0.61, 1.40] | 0.99 | 0.982 | [0.58, 1.70] |
| 3 and more | Ref | | | Ref | | |
| **Previous abortion experience** | 0.68 | 0.043 | [0.47, 0.99] | 1.41 | 0.202 | [0.83, 2.40] |
| **Level of education** | | | | | | |
| None/some non-formal | 0.98 | 0.959 | [0.46, 2.07] | 3.13 | 0.042 | [1.04, 9.43] |
| Primary (1–5) | 1.23 | 0.574 | [0.60, 2.51] | 3.77 | 0.014 | [1.31, 10.85] |
| Secondary (6–12) | 1.14 | 0.668 | [0.62, 2.12] | 1.87 | 0.198 | [0.72, 4.83] |
| More than secondary | Ref | | | Ref | | |
| **Employed** | | | | | | |
| No | 1.51 | 0.015 | [1.08, 2.10] | 1.72 | 0.016 | [1.11, 2.66] |
| Yes | Ref | | | Ref | | |
| **Caste/Ethnicity** | | | | | | |
| Brahmin/Chhetri | Ref | | | Ref | | |
| Hill Janajati | 0.71 | 0.093 | [0.48, 1.06] | 1.00 | 1.000 | [0.58, 1.71] |
| Dalit | 1.03 | 0.901 | [0.65, 1.62] | 2.01 | 0.015 | [1.14, 3.53] |
| Terai Janajai | 0.70 | 0.118 | [0.45, 1.09] | 0.75 | 0.337 | [0.42, 1.34] |
| Others | 0.38 | 0.036 | [0.16, 0.94] | 0.48 | 0.235 | [0.14, 1.62] |
| **Facility Type** | | | | | | |
| Public | Ref | | | Ref | | |
| Private/NGO clinic | 1.85 | 0.314 | [0.56, 6.11] | 0.91 | 0.812 | [0.44, 1.90] |
| **Eligibility for abortion** | | | | | | |
| No-sex selection reason alone | 9.39 | 0.000 | [3.90, 22.58] | 0.84 | 0.637 | [0.42, 1.71] |
| Yes-diagnosed physical health risk | 1.65 | 0.157 | [0.82, 3.31] | 1.05 | 0.918 | [0.38, 2.90] |
| Yes-three or more mental health conditions | 1.14 | 0.415 | [0.83, 1.57] | 0.89 | 0.580 | [0.59, 1.34] |
| Yes-rape/incest | 0.47 | 0.434 | [0.07, 3.07] | 0.84 | 0.908 | [0.05, 15.05] |
| Yes-fetal anomaly diagnosis | 0.51 | 0.029 | [0.28, 0.93] | 0.38 | 0.075 | [0.13, 1.10] |
| **Aware of legality of abortion** | | | | | | |
| Yes | Ref | | | Ref | | |
| No | 1.40 | 0.042 | [1.01, 1.92] | 0.74 | 0.154 | [0.50, 1.12] |
| To some extent | 1.72 | 0.173 | [0.79, 3.74] | 0.56 | 0.281 | [0.20, 1.60] |
| **Wealth Quintiles** | | | | | | |
| 1-lowest | 1.78 | 0.048 | [1.00, 3.15] | 4.38 | 0.000 | [1.99, 9.64] |
| 2 | 1.34 | 0.262 | [0.80, 2.22] | 2.32 | 0.025 | [1.11, 4.84] |
| 3 | 1.11 | 0.670 | [0.69, 1.79] | 2.07 | 0.042 | [1.03, 4.19] |

(*Continued*)

**Table 5.** (Continued)

| | Predictors of Denial | | | Predictors of Carrying Pregnancy to Term after Denial | | |
|---|---|---|---|---|---|---|
| | Adjusted Odds Ratio | P value | 95% Confidence Interval | Adjusted Odds Ratio | P value | 95% Confidence Interval |
| 4 | 1.21 | 0.420 | [0.76, 1.93] | 1.61 | 0.195 | [0.78, 3.29] |
| 5-highest | Ref | | | Ref | | |
| **Travel time to the clinic** | | | | | | |
| Up to 1/2 hour | Ref | | | Ref | | |
| >1/2 to 1 hour | 1.20 | 0.371 | [0.81, 1.78] | 1.27 | 0.386 | [0.74, 2.20] |
| >1 to 3 hours | 1.38 | 0.123 | [0.92, 2.09] | 1.19 | 0.536 | [0.69, 2.07] |
| >3 to 24 hours | 1.49 | 0.104 | [0.92, 2.40] | 0.65 | 0.174 | [0.35, 1.21] |

and lack empowerment in other ways, thus making it more challenging for them to insist on receiving services. This lack of empowerment could disadvantage women in their households (affecting the timing of presenting at the facility and subsequent attempts through low decision-making power) as well as at the community level (affecting their ability to negotiate at the health facility) [20]. Stigma and provider bias may also play a role in determining who can access care at the facility for women who are able to get to a facility on time. Knowledge of the legality of abortion among patients appeared to be an important facilitator of early care seeking, highlighting the importance of increasing public awareness of the availability of legal abortion services.

The majority of those denied abortions were told that it was because they were too far along in the pregnancy; while most of these women were past 10 weeks, not all were, and given that the legal limit for abortion on request is 12 weeks, many of these women met the legal criteria for abortion. Previous work has shown that many providers are not aware of the criteria for legal abortion beyond 12 weeks and do not regularly screen for eligibility before turning women away [19]. In the present study, the great majority of those who were denied the procedure beyond 12 weeks should have been deemed eligible for abortion. For those women who reported not being sure that they wanted an abortion as a reason for denial (6%), more work is needed to disentangle whether this is post-denial acceptance of the pregnancy or a change of mind. Other reasons for denial, such as the woman reporting that the provider did not do abortions (6% of denials), may represent miscommunication between the provider and patient since all study facilities are known to provide abortion services. Lack of availability of the provider (5% of all denials but 26% of those under 10 weeks) may indicate that the facility does not have medication abortion pills on hand or that the doctor/provider was not present that day (in which case only medication abortion up to 9 weeks is permitted). Reporting a lack of adequate money to pay for the services is an infrequent reason for denial (4%) but it indicates a potential problem with communication since abortions should be available without cost at public facilities and women can be referred there from the private facilities.

The finding that many of those who were denied at recruitment facilities should have been legally eligible for abortion care likely explains why so many were able to obtain abortions elsewhere after denial. Nevertheless, having to seek care at multiple facilities is confusing and presents logistical and timing challenges for patients, particularly those with resource constraints. Although most women who were denied were ultimately able to terminate the pregnancy, a lack of streamlined pathways to care increases the burdens of abortion-seeking, including travel costs, childcare needs, and lost wages, as well as the emotional and physical difficulties of remaining pregnant for longer than one desires. Such complex and inefficient pathways to

care also likely contribute to misinformation among patients regarding where and when to seek abortion in Nepal.

The finding that many women report that they want an abortion because they have enough children or need to take care of the children they already have indicates that denial of abortion services may have profound impacts on the wellbeing of children living in the household. Indeed, this is what findings from the US also suggest [21–23]. Understanding these impacts is one goal for the longitudinal data still to be collected. Finally, concerns for the physical and psychological health impacts of pregnancy, the reason for abortion-seeking for one in six women, are important and will be explored in further analyses. Maternal mortality and morbidity in Nepal are exceedingly high [24] and the consequences of not having control over the decision to carry a pregnancy to term and give birth may be dire for the wellbeing of women and their families.

Study limitations include loss to follow up of those who completed the baseline but not 6-week interview (n = 167, 9%), the possibility of social desirability bias where the 6% who said they were uncertain about wanting an abortion after being denied may have said so because they were unable to get care, and possible underreporting of sex selection as a reason for abortion to the extent it is stigmatized or people know that it is illegal to seek abortion for sex selection reasons. This study is strengthened by its large sample size, high follow-up rate, and national representation of women seeking abortion in every district of Nepal (at least in phase one of recruitment). It is also the first study to follow women over time to understand the effects of receiving versus being denied an abortion.

Despite Nepal's extensive and long-standing efforts to make abortion services legal and widely available, findings from this study show that some women in Nepal are still being denied abortions, including those closer to, but still within, the legal limit for abortion on request and those with indications for legal abortion beyond that limit. Socially and economically disadvantaged women are more likely to seek abortion care later in pregnancy, to be denied abortion care, and to carry the pregnancy to term once they have been denied. Programs and policies are needed to help ensure that all who are legally eligible to obtain abortions can; for example, by addressing potential bias, lack of knowledge, and resource capacity among providers. Such programs also should focus on comprehensive provider training about legal eligibility for abortion, medical and human resource allocation, and streamlined referral processes to ensure that all women and girls who are eligible can obtain abortions services. Additionally, empowering women (especially those that face other intersecting forms of disempowerment due to poverty or young age) as well as their family members and communities with information and resources may help women to seek abortion care earlier and obtain the services that they desire in a timely manner without having to go to multiple facilities. Future analyses from this study will focus on exploring the effects of being denied versus receiving an abortion with regard to maternal physical and mental health, socioeconomic consequences, relationships and partner violence, women's empowerment, achievement of aspirational plans, and the well-being of existing and future children.

## Acknowledgments

We would like to thank all of the field researchers who did the primary recruitment and data collection, the study sites, members of the national technical advisory committee, and the participants for their time.

## Author Contributions

**Conceptualization:** Mahesh C. Puri, Diana Greene Foster.

**Data curation:** Mahesh C. Puri, Sarah Raifman, Dev Chandra Maharjan, Chris Ahlbach.

**Formal analysis:** Mahesh C. Puri, Sarah Raifman, Sara Daniel, Chris Ahlbach, Nadia Diamond-Smith, Diana Greene Foster.

**Funding acquisition:** Diana Greene Foster.

**Methodology:** Mahesh C. Puri, Sara Daniel, Sunita Karki, Dev Chandra Maharjan, Diana Greene Foster.

**Project administration:** Mahesh C. Puri, Sunita Karki, Dev Chandra Maharjan.

**Software:** Dev Chandra Maharjan, Diana Greene Foster.

**Supervision:** Mahesh C. Puri, Sunita Karki.

**Validation:** Dev Chandra Maharjan.

**Visualization:** Sunita Karki, Dev Chandra Maharjan.

**Writing – original draft:** Mahesh C. Puri, Sarah Raifman, Chris Ahlbach, Nadia Diamond-Smith, Diana Greene Foster.

**Writing – review & editing:** Mahesh C. Puri, Sarah Raifman, Sara Daniel, Nadia Diamond-Smith, Diana Greene Foster.

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
