## [Editor Report · Decision Letter 0]

18 Jul 2022

PONE-D-22-07816Denial of legal abortion in NepalPLOS ONE

Dear Dr. Puri,

Thank you for submitting your manuscript to PLOS ONE. After careful consideration, we feel that it has merit but does not fully meet PLOS ONE’s publication criteria as it currently stands. Therefore, we invite you to submit a revised version of the manuscript that addresses the points raised during the preliminary assessment.

I have now gone through the manuscript “Denial of legal abortion in Nepal”. The authors have conducted a study on an important but neglected topic in the context of Nepal. Before the manuscript is subjected for peer review, I suggest some edits in the manuscript.

1. Please ensure that the cover letter is intended for submission of the manuscript to Plos One.

2. Please check reference 13 and 14 for their completeness. I recommend the authors cited Muluki Criminal (Code) Act 2074 (2017), Paragraph 13, Article 189 for the legislature relating to abortion in Nepal. https://www.ilo.org/dyn/natlex/docs/ELECTRONIC/106060/129899/F1095481449/NPL106060%20Npl.pdf 

3. Regarding facilities and centers providing abortion care the authors have retrieved the information from the articles cited as references 15, 16 and 17, the latest being 2019. I recommend the authors updated the manuscript with latest current figures from the primary source (Ministry of Health and Population) rather than the cross-reference. https://www.mohp.gov.np/eng/program/reproductive-maternal-health/nsas 

4. Please ensure uniformity in the reference style.

5. The reference section shows too much of self-citation. Please ensure only the required and relevant articles are cited in the context to the present study. 

We look forward to receiving your revised manuscript.

Kind regards,

Alok Atreya

Academic Editor

PLOS ONE

Journal Requirements:

This study was supported by the National Institute of Health (Grant number: A133916), and the Packard Foundation (Grant Number 132968) to Diana Greene Foster, at the University of California, San Francisco. 

No authors report competing interests.
---

## [Author Response · Author response to Decision Letter 0]

19 Aug 2022

Point-by-point response to comments/suggestions

1. Please ensure that the cover letter is intended for the submission of the manuscript to Plos One.

Response: The cover letter has been revised. 

2. Please check reference 13 and 14 for their completeness. I recommend the authors cited Muluki Criminal (Code) Act 2074 (2017), Paragraph 13, Article 189 for the legislature relating to abortion in Nepal. https://www.ilo.org/dyn/natlex/docs/ELECTRONIC/106060/129899/F1095481449/NPL106060%20Npl.pdf

Response: We have cited the most recent and relevant for this: ‘Safe Motherhood and Reproductive Health Rights Act, 2075 (2018), Chapter 4, Article 15. https://www.lawcommission.gov.np/en/wp-content/uploads/2019/07/The-Right-to-Safe-Motherhood-and-Reproductive-Health-Act-2075-2018.pdf

We have updated our references accordingly. 

3. Regarding facilities and centers providing abortion care the authors have retrieved the information from the articles cited as references 15, 16 and 17, the latest being 2019. I recommend the authors updated the manuscript with latest current figures from the primary source (Ministry of Health and Population) rather than the cross-reference. https://www.mohp.gov.np/eng/program/reproductive-maternal-health/nsas

Response: Thank you – however, the suggested government citation (MOHP website) has not been updated regularly- the latest information available from the suggested site is for between 2014/15 (FY71/72). Therefore, we have used another MOHP reference to update these numbers. After 2019/2020 the government has not updated these figures in their annual report or any other publications after 2019/2020. For this, we cited the following reference 

Ministry of Health and Population, Annual Report Department of Health Services 2076/77 (2019/2020). https://dohs.gov.np/wp-content/uploads/2021/07/DoHS-Annual-Report-FY-2076-77-for-website.pdf

4. Please ensure uniformity in the reference style.

Response: We have checked the uniformity in the reference style carefully and revised it. 

5. The reference section shows too much of self-citation. Please ensure only the required and relevant articles are cited in the context to the present study. 

Responses: These are carefully checked and avoided as much as possible. We deleted 4 references including two self-citations. Moreover, we have references only that are needed and we seem self-referential because very few other research groups are studying abortion denial in Nepal and elsewhere. 

Journal Requirements:

Response. PLoS ONE’s style has been checked while formatting our manuscript. 

Response: We have added the following sentence in the method section:

“In the case of minor under 18 years of age, consent from one biological parent (either mother or father) and assent from the girl was obtained”.

This study was supported by the National Institute of Health (Grant number: A133916), and the Packard Foundation (Grant Number 132968) to Diana Greene Foster, at the University of California, San Francisco. 

Response: We have revised the financial disclosure as “The funders had no role in study design, data collection and analysis, decision to publish, or preparation of the manuscript”. This has also mentioned in the cover letter. 

No authors report competing interests.

Response: This section has been updated as suggested. We have also mentioned this our cover letter. 

Response: Since this manuscript is from an ongoing longitudinal study – data cannot be made public. However, De-identified data used in the preparation of this manuscript will be available upon reasonable request. The entire data set will also be made public at the end of the study.

---

## [Decision Letter · Decision Letter 1]

7 Feb 2023

PONE-D-22-07816R1Denial of legal abortion in NepalPLOS ONE

Dear Dr. Puri,

Thank you for submitting your manuscript to PLOS ONE. After careful consideration, we feel that it has merit but does not fully meet PLOS ONE’s publication criteria as it currently stands. Therefore, we invite you to submit a revised version of the manuscript that addresses the points raised during the review process.

We look forward to receiving your revised manuscript.

Kind regards,

Kanchan Thapa, MPH, MPhil

Academic Editor

PLOS ONE

Journal Requirements:

Additional Editor Comments:

Dear Dr Puri,

It’s nice to read your paper and I found few issues to be addressed before the publication. Please also refers to review comments too.

The paper is well written however there is still rooms for improvement in each and every section. Few typos have been seen along with fluency of language in some parts of the paper. I have few specific comments as below:

Reorganize the abstract. I did not see how many participants were included in the study and findings in terms of statistical terms such as aOR (CI), x%(a/b) etc.. It is better to state clearly about your research design. What is the consistency for calculating wealth quantile with NDHS?

Methods:

You did not mention about the sample size and study design clearly. Please make it clear.

Not reaching till the result section, I found it is confusing about how any sampling sites. Please make it clear throughout the methodology section.

Results

Line 219- 1,841 (95.3% of eligible women), who are these eligible women and how many?

Revise the presentation of table and make them as per standards. Please differentiate between use of N and n. Make consistent throughout the tables.

I did not see any significance of each 100% and N data in tables separately.

Table 2. what does that mean Col % Col % Col %?

Describe about mixed effect in methodology section and why does it happen?

Line 256- Referencing 1- is it an appropriate referencing style?

Line 268- Make standard referring to table. Please review other paper for writing styles.

I found analytical write up in result section is compromised. Please write the result section in more analytical and interpretative way.

Found about P>z, can you please have mentioned about the use of P>z and its significant in methodology section.

Discussion:

More analytical presentation of result is required along with comparing with other literature.

Reviewers' comments:

Reviewer's Responses to Questions

**Comments to the Author**

1. If the authors have adequately addressed your comments raised in a previous round of review and you feel that this manuscript is now acceptable for publication, you may indicate that here to bypass the “Comments to the Author” section, enter your conflict of interest statement in the “Confidential to Editor” section, and submit your "Accept" recommendation.

Reviewer #1: (No Response)

Reviewer #2: (No Response)

2. Is the manuscript technically sound, and do the data support the conclusions?

Reviewer #1: Yes

Reviewer #2: Yes

3. Has the statistical analysis been performed appropriately and rigorously? 

Reviewer #1: Yes

Reviewer #2: Yes

4. Have the authors made all data underlying the findings in their manuscript fully available?

Reviewer #1: Yes

Reviewer #2: Yes

5. Is the manuscript presented in an intelligible fashion and written in standard English?

Reviewer #1: No

Reviewer #2: Yes

6. Review Comments to the Author

Reviewer #1: There is a need to get the paper rechecked by a native English speaker.

Avoid using "our" in the write-up for example, in line 43 replace "our" with "a".

In the background section, there is a need to add relevant literature from international studies. This shall be done with an intention to provide a conceptual framework for the regression model used in this study. In other words, the set of independent variables should be clearly mapped with relevant literature (for example, which studies back the inclusion of wealth quintile as an independent variable).

Please add some limitations, and avenues for future work. The paper ends abruptly; please add some policy recommendations that can be linked directly with the analysis conducted in the current study.

Reviewer #2: The objective of the study is clearly stated by the authors. The data is organized in an effective manner and the statistical analysis is conducted with a high level of proficiency. As a result, the paper has a significant impact on the healthcare system of Nepal.

7. PLOS authors have the option to publish the peer review history of their article (what does this mean?). If published, this will include your full peer review and any attached files.

Reviewer #1: **Yes: **Dr. Ayesha Nazuk

Reviewer #2: **Yes: **Laxman Datt Bhatt

---

## [Author Response · Author response to Decision Letter 1]

22 Feb 2023

A point-by-point response to the Editor and Reviewers’ comments/suggestions

Journal Requirements:

Additional Editor Comments:

Dear Dr Puri,

It’s nice to read your paper and I found few issues to be addressed before the publication. Please also refers to review comments too.

The paper is well written however there is still rooms for improvement in each and every section. Few typos have been seen along with fluency of language in some parts of the paper. 

Response: Thank you for the suggestions. We have reviewed the paper in its entirety, revised the language, and fixed typos. Two co-authors of this paper who are native English speakers have carefully checked fluency in language and grammar. 

I have few specific comments as below:

Reorganize the abstract. I did not see how many participants were included in the study and findings in terms of statistical terms such as aOR (CI), x%(a/b) etc.. It is better to state clearly about your research design. What is the consistency for calculating wealth quantile with NDHS?

Response: Thank you for these suggestions. We have added detail on sample numbers and descriptive findings and reorganized the abstract. It is challenging to add aORs for all of the significant predictors mentioned in the abstract; there are 3 separate models (for gestational age at presentation, denial of care, and birth subsequent to denial) and for each of the models there are several significant coefficients for levels of wealth, education, age, and other variables. Given space limitations and for clarity, we have decided not to include all of these aORs in the abstract. 

Methods:

You did not mention about the sample size and study design clearly. Please make it clear.

Not reaching till the result section, I found it is confusing about how any sampling sites. Please make it clear throughout the methodology section.

Response: We clarified in the second paragraph of the methods section that we recruited participants from a total of 22 sites, including 14 original sites and 8 additional sites. 

Results

Line 219- 1,841 (95.3% of eligible women), who are these eligible women and how many?

Response: 8,856 women were screened. We have clarified that 1,925 of those (21.8%) were eligible. Of the 1,925 eligible, 1,835 (95.3%) consented to participate and completed a baseline interview. The final analytic sample included 1668 participants. This should help clarify why Table 1 is based on data from the 1,835 who completed a baseline interview. 

Revise the presentation of table and make them as per standards. Please differentiate between use of N and n. Make consistent throughout the tables. I did not see any significance of each 100% and N data in tables separately

Response: We provide frequencies and percentages for the total (N) sample and percentages for the subgroups <10 weeks gestation and >=10 weeks/Don’t know. We removed unnecessary columns. 

Table 2. what does that mean Col % Col % Col %?

Response: We have simplified the table heading to include simply “%” instead of “Col %”. We have removed this from all tables. Please note that Table 4 has missingness.

Describe about mixed effect in methodology section and why does it happen?

Response: In the methods section, we have clarified the following: “we used bivariable and multivariable mixed-effects logistic models accounting for clustering at the facility level, given that patient characteristics and service provision protocols including denial of care may be more similar within a given facility. 

Line 256- Referencing 1- is it an appropriate referencing style?

Response: This was a foot note – now we have deleted this. 

Line 268- Make standard referring to table. Please review other paper for writing styles.

Response: We have made this correction.

I found analytical write up in result section is compromised. Please write the result section in more analytical and interpretative way.

Response: Thank you for your suggestion. We are trying to both communicate the quantitative findings while emphasizing the policy importance.

Found about P>z, can you please have mentioned about the use of P>z and its significant in methodology section.

Response: We changed the description to “P value”

Discussion:

More analytical presentation of result is required along with comparing with other literature.

Response: Very few studies have been conducted in this area to compare our study findings. However, we have added a few more references and compared the results where available. 

6. Review Comments to the Author

Reviewer #1: There is a need to get the paper rechecked by a native English speaker.

Response: The manuscript was carefully re-read by the English speaking co-authors and edits were made.

Avoid using "our" in the write-up for example, in line 43 replace "our" with "a".

Response: We have removed all the “our”s that we could. One remained, referencing our previous work.

In the background section, there is a need to add relevant literature from international studies. This shall be done with an intention to provide a conceptual framework for the regression model used in this study. In other words, the set of independent variables should be clearly mapped with relevant literature (for example, which studies back the inclusion of wealth quintile as an independent variable).

Response: We appreciate this comment. In our design of the study, we relied on our previous studies both in the United States and Nepal. This expertise did not emerge from a conceptual framework but from empirical investigations.

Please add some limitations, and avenues for future work. The paper ends abruptly; please add some policy recommendations that can be linked directly with the analysis conducted in the current study.

Response: Thank you for this suggestion. We have added some limitations and strengths of the study (see lines 436-444). Lines 451-460 are about policy/program recommendations that have emerged from our study findings. We have added few sentences about future work (see lines 461-464). 

Reviewer #2: The objective of the study is clearly stated by the authors. The data is organized in an effective manner and the statistical analysis is conducted with a high level of proficiency. As a result, the paper has a significant impact on the healthcare system of Nepal.

Response: Thank you for this comment.

---

## [Editor Report · Decision Letter 2]

27 Feb 2023

Denial of legal abortion in Nepal

PONE-D-22-07816R2

Dear Dr. Puri,

We’re pleased to inform you that your manuscript has been judged scientifically suitable for publication and will be formally accepted for publication once it meets all outstanding technical requirements.

Kind regards,

Kanchan Thapa, MPH, MPhil

Academic Editor

PLOS ONE

Additional Editor Comments (optional):

Dear Dr Puri and Team,

Thank you for revising the paper and preparing as the comments from reviewers and editor. I hope the paper would add more value in the field of legalization of abortion in Nepal.

Best- Kanchan
---

## [Editor Report · Acceptance letter]

13 Mar 2023

PONE-D-22-07816R2 

Denial of legal abortion in Nepal 

Dear Dr. Puri:

I'm pleased to inform you that your manuscript has been deemed suitable for publication in PLOS ONE. Congratulations! Your manuscript is now with our production department. 

Kind regards, 

on behalf of

Mr. Kanchan Thapa 

Academic Editor

PLOS ONE